# Diagnostic Value of the TpTe Interval in Children with Ventricular Arrhythmias

**DOI:** 10.3390/ijerph182212194

**Published:** 2021-11-20

**Authors:** Joanna Jaromin, Grażyna Markiewicz-Łoskot, Lesław Szydłowski, Agnieszka Kulawik

**Affiliations:** 1Department of Nursing and Social Medical Problems, Faculty of Health Sciences in Katowice, Medical University of Silesia, 40-752 Katowice, Poland; mic54@o2.pl; 2Department of Pediatric Cardiology, Faculty of Medical Sciences in Katowice, Medical University in Silesia, 40-752 Katowice, Poland; szydlowskil@interia.pl; 3Faculty of Science and Technology, University of Silesia in Katowice, Bankowa 14, 40-007 Katowice, Poland; agnieszka.kulawik@us.edu.pl

**Keywords:** children, electrocardiography, ventricular repolarization, ventricular arrhythmias, TpTe interval

## Abstract

Background: The changes in the period of ventricular repolarization, i.e., QT interval, QTp (Q-Tpeak) and TpTe interval (Tpeak–Tend), make it possible to assess the electrical instability of the heart muscle, which may lead to the development of life-threatening ventricular arrhythmia. The aim of the study was to determine and evaluate the use of differences in T-wave morphology and durations of repolarization period parameters (QT, TpTe) in resting ECGs for children with ventricular arrhythmias. Methods: The retrospective analysis was made of the disease histories of 80 examined children with resting ECGs, which were admitted to the Children’s Cardiology Department. The study group consisted of 46 children aged 4 to 18 with ventricular arrhythmias and the control group consisted of 34 healthy children between 4 and 18 years of age, with no arrhythmias. Results: The duration of the TpTe interval was significantly (*p* < 0.001) longer in the group of children with ventricular arrhythmia with abnormal T-wave (bactrian/bifid, humid/biphasic) compared to the TpTe interval in children with ventricular arrhythmia with the normal repolarization period. The duration of the TpTe (*p* < 0.001), QTcB (*p* < 0.001) and QTcF (*p* < 0.001) intervals were significantly longer in the group of children with ventricular arrhythmias and with abnormal T-wave compared to the values of the TpTe, QTcB, and QTcF intervals of the control group with normal morphology of the repolarization period. Only the duration of the TpTe interval was significantly (*p* = 0.020) longer in the group of children with ventricular arrhythmia without clinical symptoms. Conclusions: Children with benign ventricular arrhythmias recorded on a standard ECG with prolonged TpTe and QT intervals and abnormal T-wave morphology require systematic and frequent cardiac check up with long term ECG recordings due to the possibility of future more severe ventricular arrhythmias. Further follow-up studies in even larger groups of patients are necessary to confirm the values of these repolarization parameters in clinical practice.

## 1. Introduction

Arrhythmias are an important clinical problem in the developmental population and a common cause of hospitalization. Their clinical course is most often asymptomatic, with no tangible cardiovascular abnormalities. Among the healthy population, premature ventricular beats in the standard ECG may be recorded in 0.3–2.2% of newborns, infants, and children with normal circulatory system [1]. About 50% of children and adolescents with mild ventricular arrhythmias (spontaneously disappearing in about 48–65% of children), arrhythmia is of unknown etiology without coexisting overt, organic, or primarily electrophysiological heart disease [2]. The severity of ventricular arrhythmias can be manifested in chest pain, rapid heartbeat, dizziness, blurred vision (‘darkness’ in front of the eyes), fainting and loss of consciousness, and sudden cardiac arrest [3,4,5,6]. The analysis of resting electrocardiographic and/or 24 h ECG by Holter’s method is necessary in the diagnosis of arrhythmias as a cause of syncope, circulatory failure, as well as in the assessment of the risk of sudden cardiac death [7,8,9,10]. Electrocardiographic parameters of the repolarization period: QT interval, QTp (Q-Tpeak), and especially the TpTe interval (Tpeak–Tend), which is the final part of the T-wave, is an indicator of electrical myocardial instability (heterogeneity of repolarization), the increase of which, with the change of the wave shape T (bactrian/bifid, humid/biphasic) may predispose to life-threatening arrhythmias. In the literature the TpTe interval seems to be a more sensitive indicator of arrhythmogenesis compared to the standard QT interval [1,11,12,13,14]. The aim of the study was to determine in the standard ECG electrocardiographic values of repolarization period parameters, i.e., the TpTe interval (Tpeak–Tend) and QT interval, corrected Bazett (QTcB) and Fridericia (QTcF) formulas with the assessment of T-wave morphology in a group of children with ventricular arrhythmias and in children in the control group, as well as the assessment of the possibility of using differences in T-wave morphology and durations of repolarization period parameters (QT, TpTe) determined in the standard ECG in the examined groups of children in predicting the risk of a more serious arrhythmia.

## 2. Materials and Methods

The retrospective analysis was carried out on the histories of examined children with electrocardiographic records (12-lead ECG), which were performed after each of the children was admitted to the Children’s Cardiology Department. The information included in the medical records of younger children (physical and family history) was obtained from their parents or legal guardians.

### 2.1. The Studied Subjects

A group of 80 children were included in the study. The study group consisted of 46 children (21 girls and 25 boys) aged 4 to 18 years (average age 13 ± 6 years) with ventricular arrhythmias of unknown etiology which were recorded in standard ECG in 12 children (26%) or in 24 h Holter ECG monitoring in 34 children (74%).

The control group consisted of 34 healthy children (volunteers) (22 girls and 12 boys) aged 4 to 18 years (average age 13 ± 3 years) without confirmed arrhythmias. All the children from the study group and the control group were hospitalized in the Children’s Cardiology Department of the Independent Public Clinical Hospital No. 6 of the Silesian Medical University in Katowice—Ligota (Upper Silesian Children’s Health Centre of John Paul II). 

None of the 80 children had a positive sudden cardiac death or structural heart disease in physical examination and in echocardiography using the Doppler method as well as no arrhythmias during exercise stress (Bruce protocol). 

Inclusion criteria in the group of examined children: ventricular arrhythmias in standard ECG and in the 24 h ECG Holter method, the history of arrhythmia diagnosed, normal physical examination result apart from periodically irregular heart rhythm, without signs of infection or structural features of the heart defect, negative family history regarding structural and electrophysiological abnormalities of the heart, episodes of sudden cardiac death in the child’s relatives before 30 years of age (sudden infant deaths, drowning, car accidents in unexplained circumstances), normal echocardiographic result, correct laboratory results excluding the presence of inflammation and ionic disorders (K, Ca, Mg).

Criteria for inclusion in the control group: no history of complaints suggesting cardiac arrhythmias with no cardiac arrhythmias on the ECG and the 24 h ECG Holter method, negative family history regarding structural and electrophysiological abnormalities of the heart, no episodes of sudden cardiac death in relatives of a child before 30 years of age, correct physical examination result, with particular emphasis on the state of the circulatory system, without signs of inflammation or structural features of the heart defect with normal echocardiographic result, correct laboratory results excluding the presence of inflammation and ionic disorders (K, Ca, Mg).

Exclusion criteria for testing: extra-sinus rhythm on the standard ECG and 24 h Holter ECG, features of sick sinus syndrome, atrioventricular and intraventricular conduction disorders in the right and left branches of the bundle in standard ECG and the 24 h Holter ECG recording, diagnosis of structural heart defect (including cardiomyopathies), inflammation of the heart muscle, abnormal electrolyte levels in blood laboratory tests (K, Ca, Mg), the use of drugs that change the repolarization period, the presence of diseases that change the repolarization period (connective tissue diseases, neurological and endocrine diseases), lack of consent of the patient or their legal guardian.

#### Electrocardiography

Standard ECG recordings were performed in the supine position using the recorder model AT2 plus Schiller AG. Baar, Switzerland. Measurements were made manually in the fifth pre-cardiac lead (V5) in 12-lead ECGs. The analysed measurements were the average of three consecutive QRS-T evolutions at a paper travel of 50 mm/s and a standard feature amplitude of 1 mV = 1 cm. The analysis of the repolarization period evaluated the duration of individual repolarization parameters as well as the amplitude and shape of the T-wave (bactrian/bifid, humid/biphasic). From three consecutive QRS-T evolutions, RR intervals, total repolarization duration—QT interval, and TpTe interval (TpeakTend) were determined and recorded enlarged with a magnifying glass using a distance stepper. The total QT repolarization period was determined from the beginning of the Q wave to the end of the T-wave, defined as the place of return of the descending arm of the T-wave to the isoelectric line excluding U wave. The period of late repolarization TpTe (TpeakTend) was calculated from the top of the T-wave to the end of the T-wave [15,16]. The corrected duration of the QT interval in relation to the heart rate (QTcB) was calculated according to the Bazett formula (QT/√RR) and Fridericia (QT/RR1/3) [17,18]. In the case of two T-waves peaks, the first peak of the T-wave was taken into account in measurements [19,20,21]. All measurements were analyzed blindly by two independent investigators with uniformity testing without access to results obtained and clinical data. The highest quality ECGs were analyzed without network disturbances with exclusion flat T-wave.

### 2.2. Statistical Analysis

The statistical analysis is based on statistical inference as a consequence of hypothesis verification. In the analysis the significant level was assumed to be 0.05. We verify the hypotheses according to the obtained *p*-value (“*p*” for short) for a statistical test. If the *p*-value is greater than or equal to 0.05, then there is no reason to reject the null hypothesis at a significance level of 0.05, which usually says that there are no statistically significant differences. When the *p*-value is less than 0.05, we reject the null hypothesis at the significance level of 0.05 and conclude that there are statistically significant differences.

Both parametric and non-parametric hypotheses were considered in the analysis. Most often, non-parametric hypotheses are applied when the assumption that the distribution of the examined feature is normal is not met. For each considered feature in all studied populations, the normality of the distribution was checked using the Shapiro–Wilk test. Depending on the obtained results, a parametric or non-parametric test was used. In the case of comparing two independent samples, the parametric Student’s t-test or the non-parametric Mann–Whitney U test was used. 

The results of the statistical tests are presented in the tables. In the case of parametric tests, the following were given: arithmetic mean and standard deviation calculated for the considered samples (groups of children), as well as the number of samples (row N) and the name of the test used together with the *p*-value. In the case of non-parametric tests, the following were given: the sum of ranks and the mean of the ranks obtained for the considered samples (groups of children), as well as the sample size (row N) and the name of the test used together with the *p*-value.

The boxplots were also used in the presentation of the results. From such charts one can read the basic measures of descriptive statistics. And more specifically: the median—the horizontal line in the middle of the box, the first quartile—the lower base of the box, the third quartile—the top of the box, the minimum–the level of the lower whisker, the maximum—the level of the upper whisker.

The calculations and the graphs used in the article were made in the Statistica software (version 13.1, StatSoft, TIBCO Software Inc., Palo Alto, California, United States).

### 2.3. Ethical Approval

The research project was approved by the Bioethical Committee of the Medical University of Silesia in Katowice in 2015, No. KNW/0022/KB/238/15.

## 3. Results

In children with a history of ventricular arrhythmia, the most commonly reported clinical manifestations were poor exercise tolerance (*n* = 8, 36.4%), stabbing chest pain (*n* = 5, 22.7%), chest twinge (*n* = 5, 22.7%) and fainting (*n* = 4, 18.2%). There were no complaints in the control group of healthy children (Table 1). In the 12-lead ECG performed at admission, 12 (27%) children with ventricular arrhythmia had single premature ventricular beats (ventricular extrasystoles) (Table 1). In 22% of children with ventricular arrhythmia, abnormal T-waves in lead V4, V5, and V6 were observed in the standard ECG: bactrian/bifid T-waves in five children (11%) and humid/biphasic T-waves in five children (11%) (Table 1).

All ventricular extrasystoles originated from the left ventricle (with right bundle branch block (RBBB) morphology).

Due to the number of registered extrasystoles in 24 h Holter ECG monitoring, 4 subgroups were distinguished in the entire study group: (1) less than 1000 a day, (2) 1000–10,000 per day, (3) over 10,000 a day, (4) recorded tachycardia.

In the group of children with ventricular arrhythmia, the most frequently (*n* = 39, 85%) that children recorded extrasystoles was in the number of 1000 to more than 10,000 per day (Table 2).

Of all the compared repolarization parameters, only the duration of the TpTe interval was significantly (*p* = 0.033) longer in the group of children with ventricular arrhythmia with an abnormal T-wave (bactrian, humid) relative to the TpTe interval in children with ventricular arrhythmia with the normal repolarization period (Table 3 and Table 4, Figure 1, Figure 2, Figure 3, Figure 4 and Figure 5).

The duration of the TpTe (*p* < 0.001), QTcB (*p* < 0.001) and QTcF (*p* < 0.001) intervals were significantly longer in the group of children with ventricular arrhythmias and with T-wave abnormal (bactrian, humid/biphasic) compared to the values of the TpTe, QTcB, and QTcF intervals of the control group with normal morphology of the repolarization period (Table 5 and Table 6, Figure 6, Figure 7, Figure 8, Figure 9 and Figure 10).

Compared to the duration of the QT, QTp, QTcB, QTcF, and TpTe intervals, only the duration of the TpTe interval was significantly (*p* = 0.020) longer in the group of children with ventricular arrhythmia without clinical symptoms (Table 7 and Table 8, Figure 11, Figure 12, Figure 13, Figure 14 and Figure 15).

The duration of the TpTe (*p* < 0.001) interval was significantly longer in the group of 26% children with ventricular extrasystoles recorded on standard ECG compared to the value of the TpTe interval in the rest of children with ventricular arrhythmias on 24 h Holter ECG monitoring. Similarly, the number of ventricular extrasystoles also in bigemini and trigemini form was significantly higher in the children with ventricular arrhythmias recorded on standard ECG then in the rest of study group with ventricular arrhythmias (Table 9). 

## 4. Discussion

In our own study of 46 children with ventricular arrhythmias the most commonly reported clinical manifestations were poor exercise tolerance (36.4%), stinging pain in the chest (22.7%), chest twinge (22.7%), fainting (18.2%), and palpitations in 9.1% of children. Similarly, in the studies by West et al. in 219 children with ventricular arrhythmia without clinically concomitant overt heart disease, chest pain (9%), palpitations (7%), and syncope (3%) were the most commonly reported, and at other authors’ in a group of 59 children the feeling of irregular heartbeat prevailed (13.6%) [22,23]. The importance of careful collection of anamneses of children with ventricular arrhythmia is confirmed by the study by Harris et al. in which 82% of children with ventricular arrhythmia reported no symptoms, but 46% of patients in the surveyed group asked again about specific symptoms confirmed their occurrence [24]. In our study the average age of the study group, 13 ± 6 years, was the highest compared to the average age of children in the reports cited above, which could explain the discrepancy in the frequency of the reported arrhythmia symptoms. The younger age of the examined patients with difficulties in describing the ailments may be important in perceiving and interpreting symptoms which is why it is so important to complete the interview with the parents and carers of sick children. In our study only the duration of the analyzed TpTe interval was significant in children with clinical symptoms from those in asymptomatic patients in the group of children with ventricular arrhythmia. Similarly, in the Cetin et al. it is reported that in the course of psoriasis with abnormal electrocardiogram there were significant differences in the duration of the TpTe interval when comparing patients with reported symptoms to asymptomatic ones [25].

In our own research 22% of the children with ventricular arrhythmia observed in precordial ECG leads (V4–V6) abnormal T-waves (bactrian/bifid or humid/biphasic) with a significantly prolonged duration of the TpTe interval compared to the TpTe value in children with ventricular arrhythmia and normal period repolarization and with a prolonged duration in relation to the TpTe, QTcB, and QTcF values in children in the control group. The presence of abnormal T-waves with prolonged TpTe interval, a diagnostic indicator of arrhythmogenesis [26], burdens examined patients with the possibility of life-threatening ventricular arrhythmia [27,28]. Abnormal T-waves are a characteristic morphological disorder of the repolarization period in the prolonged QT syndrome (LQTS). In 66–88% of carriers of the LQTS2 genotype the T-wave in the V4–V6 lead is most often flat and/or bifid [29], with 6% of symptomatic children with LQTS and 12–30% of asymptomatic carriers, the duration of the QT interval may be normal, below 440 ms [30]. Lehmann et al. in a study involving 254 patients from 13 families diagnosed with LQTS found a two-humped T-wave ECG in 53% of subjects with prolonged QT interval duration (QTc > 470 ms) in 27% of patients with QTc from 420 to 460 ms and in 5% of patients with normal QT interval (QTc < 410 ms) [31]. Malfatto et al., in 53 patients with LQTS, registered biphasic T-waves in 62% of patients while in the 53-person control group biphasic T-waves were present in 15% of patients [32]. Morphological analysis of abnormal T-waves with QT and TpTe interval analysis may be helpful in identifying patients at increased risk of severe ventricular arrhythmia leading to syncope and sudden cardiac death [27].

The increased duration of the QT repolarization period, especially its final phase, from the top to the end of the 262 T-wave TpTe interval is associated with an increased risk of symptomatic ventricular arrhythmias [33,34,35,36,37,38].

In the study of Wolk et al., in 6 out of 13 adult patients with induced ventricular arrhythmia the duration of the TpTe interval was longer compared to the TpTe interval of 7 patients without provoked arrhythmia [39]. Similarly, studies by Lubiński et al. found a longer duration of the TpTe interval in 18 patients with ischemic heart disease and recorded ventricular tachycardia compared to 16 patients who had a myocardial infarction without ventricular arrhythmia [40]. Lubiński et al. (7 patients with LQTS vs 10 healthy patients) and Viitasalo et al. (59 patients with LQTS vs 31 health patients) reported a significantly longer TpTe interval in patients with arrhythmogenic long QT syndrome (LQTS) compared to healthy patients [41,42]. Topilski et al. found an increased risk of polymorphic ventricular tachycardia (TdP) in patients with prolonged TpTe and QTcB intervals, especially those associated with bipolar T-waves in the course of acquired LQTS syndrome due to bradyarrhythmia [28]. Yamaguchi et al. (12 patients with LQTS and TdP vs 15 patients with LQTS without TdP) showed a longer TpTe interval in patients with LQTS with polymorphic ’torsade de pointes’ compared to patients with LQTS without registered TdP tachycardia [27]. Shimizu et al. found that the prolonged duration of the TpTe interval, more than the QTc interval, predicts sudden cardiac death in patients with hypertrophic cardiomyopathy [43].

The evaluation of repolarization period parameters, especially the TpTe interval, is useful in identifying patients at high risk of serious cardiac arrhythmias also in other clinical conditions such as short QT syndrome [44], long QT syndrome (LQTS) [45], hypertension [46,47], arrhythmogenic right ventricular cardiomyopathy [48], and in competitive sports [49]. The results of our own research are comparable to the opinions of many authors talking about the desirability of assessment in a 12-lead ECG, not only of the duration of the QT interval, but also the TpTe interval of the recognized new arrhythmogenesis index, which is considered a more sensitive prognostic parameter of ventricular arrhythmia compared to the one traditionally determined by QT interval [40,41,42].

The above studies of the parameters of the repolarization period were experimental or carried out in adult patients, there are few studies in children.

In our study the duration of the TpTe, QTcF and QTcB interval was significantly longer in children with ventricular arrhythmia compared to the values of these parameters in the healthy children from the control group.

Additionaly, in 22% of the study group with ventricular arrhythmia, abnormal T-waves with prolonged duration of the TpTe interval compared to the TpTe value in children with ventricular arrhythmia and normal T-wave were observed. Moreover, recording of benign ventricular arrhythmias in the standard ECG in 26% of examined children with an increased number of extrasystoles also in form of bigemini and trigemini on Holter ECG, and above all with a significant prolongation of TpTe interval with a potential pro-arrhytmic effect, may be a useful prognostic tool in assessing the risk of serious and even life-threatening arrhythmias in the future.

These are novel findings in children with ventricular arrhythmias of unknown etiology with the analysis of parameters of the repolarization period and no similar data have been found in available literature to date.

On the basis of our work, we wanted to show how important it is in everyday clinical practice to determine the parameters of the repolarization period in the ECG standard, especially in children with diagnosed ventricular arrhythmia because detected prolongation of the TpTe and QT intervals with the abnormal T-wave morphology on a standard ECG may be useful for identifying children more predisposed to severe ventricular arrhythmias in the future.

## 5. Conclusions

Children with benign ventricular arrhythmias recorded on a standard ECG with prolonged TpTe and QT intervals and abnormal T wave morphology require systematic and frequent cardiac check up with long term ECG recordings due to the possibility of future more severe ventricular arrhythmias. Further follow-up studies in even larger groups of patients are necessary to confirm the values of these repolarization parameters in clinical practice.

## Figures and Tables

**Figure 1 ijerph-18-12194-f001:**
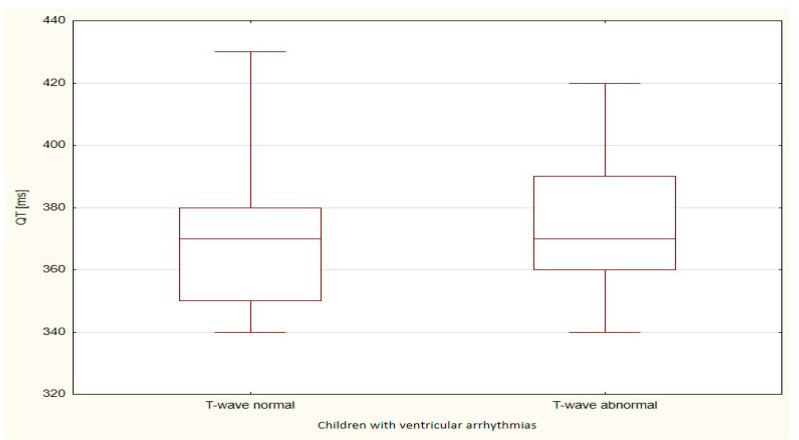
QT parameter in children with ventricular arrhythmia depending on the T-wave morphology.

**Figure 2 ijerph-18-12194-f002:**
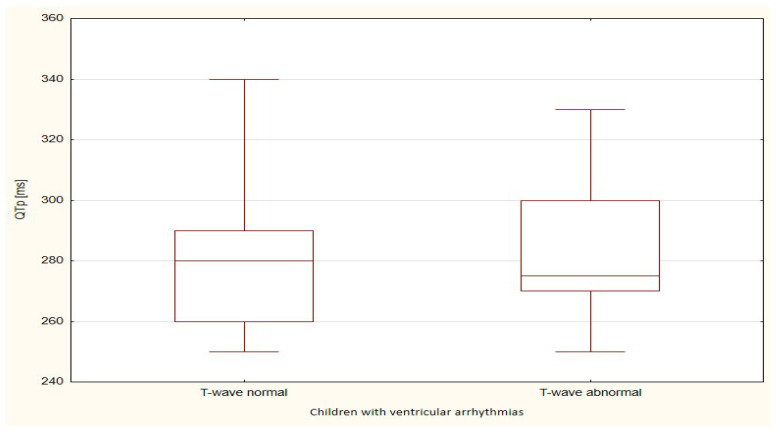
QTp parameter in children with ventricular arrhythmia depending on the T-wave morphology.

**Figure 3 ijerph-18-12194-f003:**
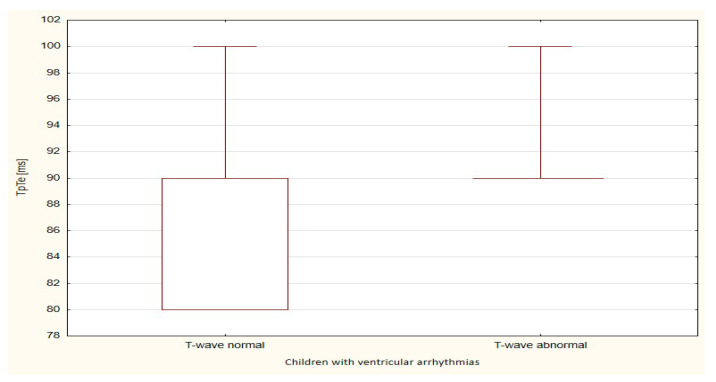
TpTe parameter in children with ventricular arrhythmia depending on the T-wave morphology.

**Figure 4 ijerph-18-12194-f004:**
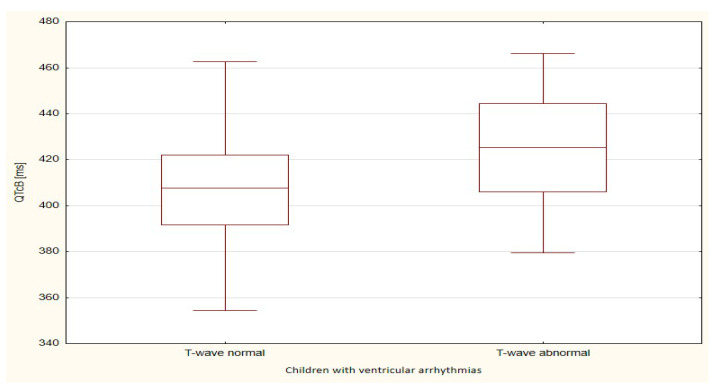
QTcB parameter in children with ventricular arrhythmia depending on the T-wave morphology.

**Figure 5 ijerph-18-12194-f005:**
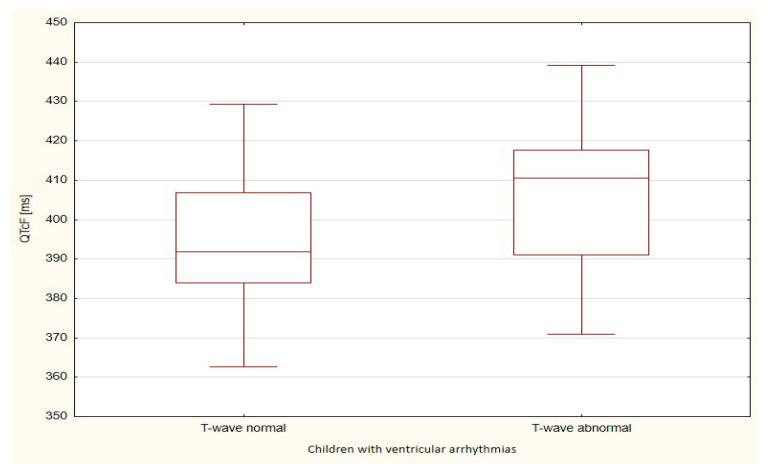
QTcF parameter in children with ventricular arrhythmia depending on the T-wave morphology.

**Figure 6 ijerph-18-12194-f006:**
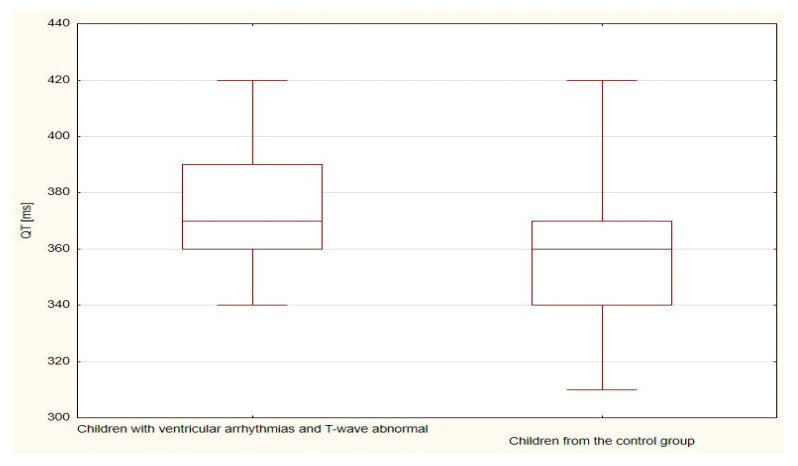
QT parameter in children with ventricular arrhythmia and T-wave abnormal and children from the control group.

**Figure 7 ijerph-18-12194-f007:**
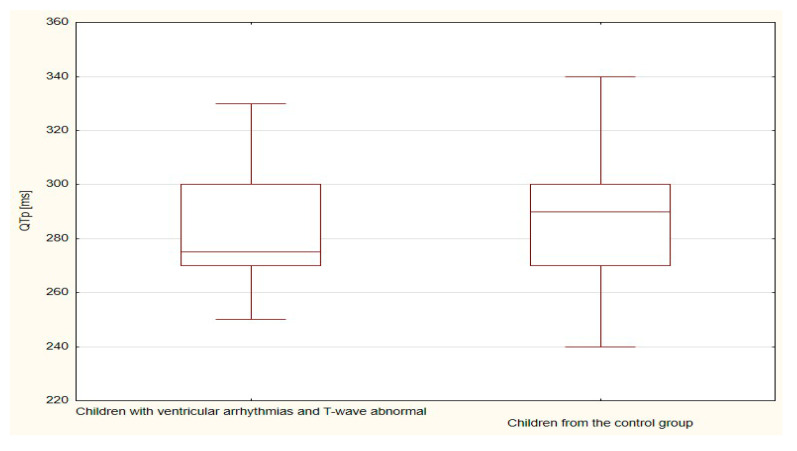
QTp parameter in children with ventricular arrhythmia and T-wave abnormal and children from the control group.

**Figure 8 ijerph-18-12194-f008:**
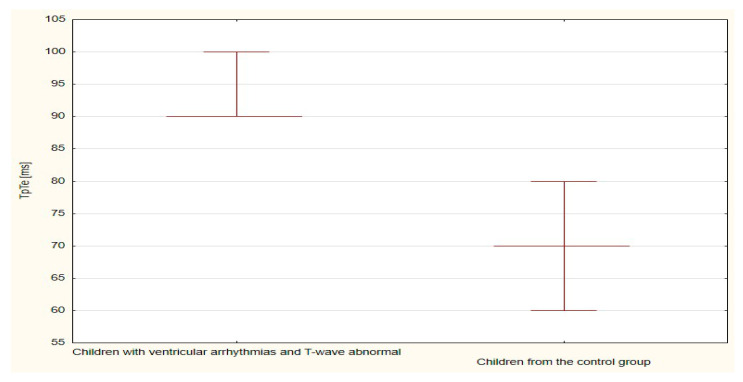
TpTe parameter in children with ventricular arrhythmia and T-wave abnormal and children from the control group.

**Figure 9 ijerph-18-12194-f009:**
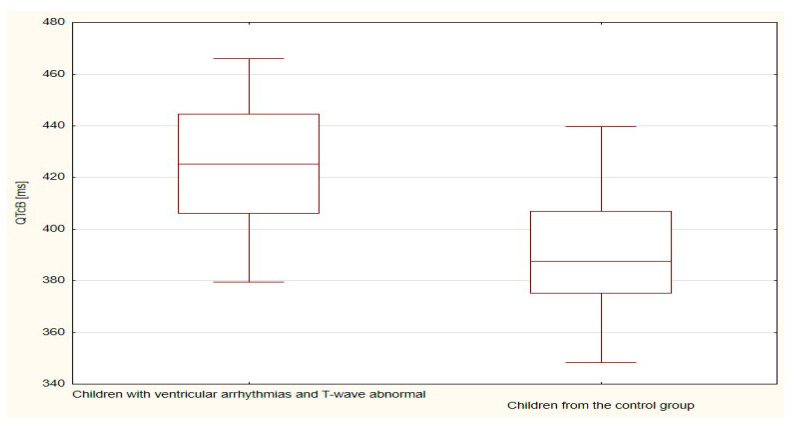
QTcB parameter in children with ventricular arrhythmia and T-wave abnormal and children from the control group.

**Figure 10 ijerph-18-12194-f010:**
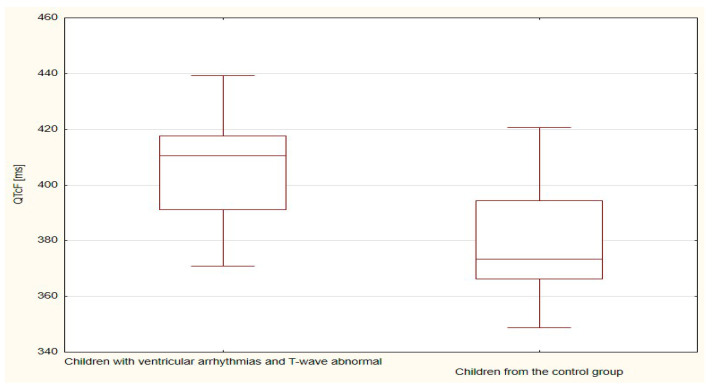
QTcF parameter in children with ventricular arrhythmia and T-wave abnormal and children from the control group.

**Figure 11 ijerph-18-12194-f011:**
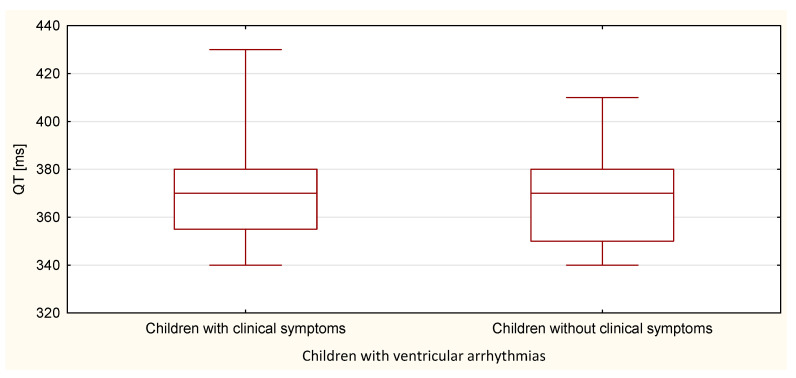
QT parameter in the group of children with ventricular arrhythmia with clinical symptoms and without them.

**Figure 12 ijerph-18-12194-f012:**
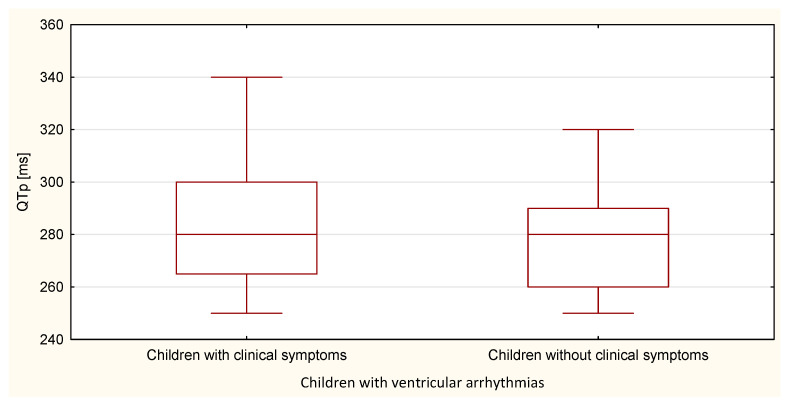
QTp parameter in the group of children with ventricular arrhythmia with clinical symptoms and without them.

**Figure 13 ijerph-18-12194-f013:**
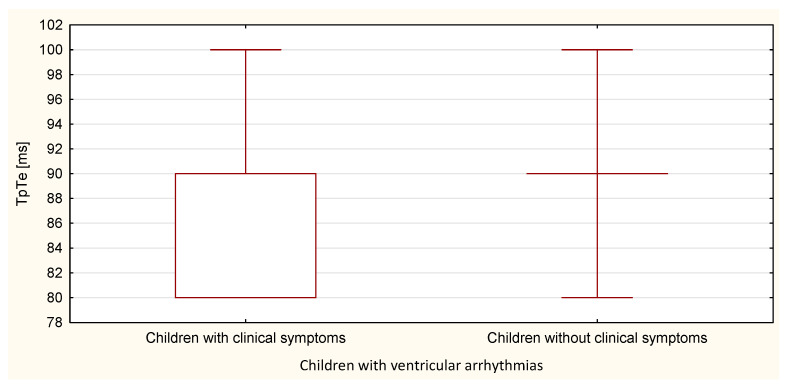
TpTe parameter in the group of children with ventricular arrhythmia with clinical symptoms and without them.

**Figure 14 ijerph-18-12194-f014:**
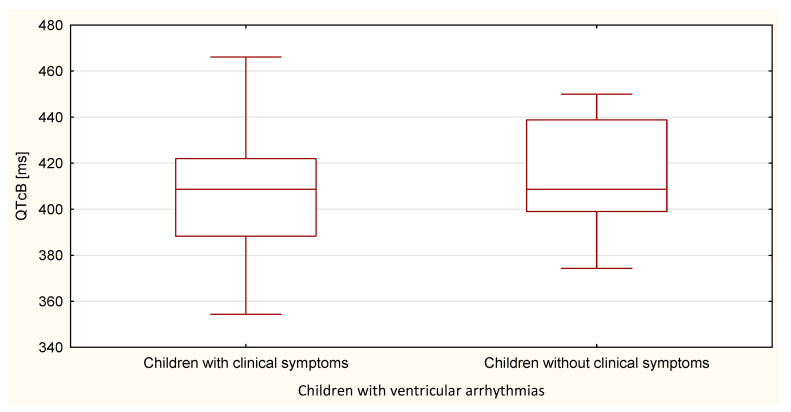
QTcB parameter in the group of children with ventricular arrhythmia with clinical symptoms and without them.

**Figure 15 ijerph-18-12194-f015:**
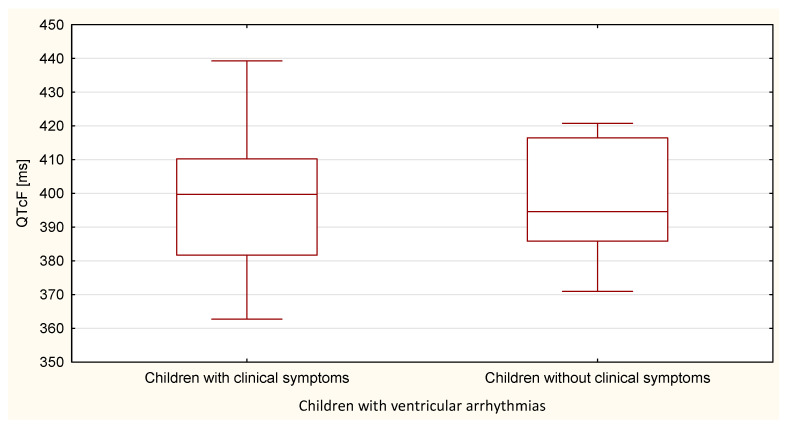
QTcF parameter in the group of children with ventricular arrhythmia with clinical symptoms and without them.

**Table 1 ijerph-18-12194-t001:** The summary of clinical observations of children with ventricular arrhythmias and healthy children of the control group.

	Total Cohort (*n* = 80)	Children with Ventricular Arrhythmias (*n* = 46)	Control Group of Healthy Children (*n* = 34)
Age (median)	13.4	13.6	13.3
Female, *n (%)*	43 (53.7%)	21 (46%)	22 (65%)
Male, *n (%)*	37 (46.3%)	25 (54%)	12 (35%)
Complaints			
Palpitations, *n (%)*	2 (2.5%)	2 (9.1%)	0
Chest twinge, *n (%)*	5 (6.2%)	5 (22.7%)	0
Chest pain, *n (%)*	1 (1.2%)	1 (4.5%)	0
Fainting, *n (%)*	4 (5%)	4 (18.2%)	0
Worse exercise tolerance, *n (%)*	8 (10%)	8 (36.4%)	0
Syncope, *n (%)*	5 (6.2%)	5 (22.7%)	0
Headache and dizziness, *n (%)*	2 (2.5%)	2 (9.1%)	0
Feeling of irregular heart-beat, *n (%)*	1 (1.2%)	1 (4.5%)	0
Ventricular extrasystoles in ECG, *n (%)*	12 (15%)	12 (26%)	0
T-wave morphology			
Bactrian, *n (%)*	5 (6.2%)	5 (11%)	0
Humid, *n (%)*	5 (6.2%)	5 (11%)	0

**Table 2 ijerph-18-12194-t002:** Characteristics of the study group with regard to extrasystoles in 24 h Holter ECG monitoring.

Number of Extrasystoles	Children with Ventricular Arrhythmias (*n* = 46)	Control Group of Healthy Children (*n* = 34)
Less than 1000 a Day	7 (15.26%)	0
1000–10,000 per Day	19 (41.30%)	0
Over 10,000 a Day	20 (43.44%)	0
Tachycardia	0	0
Total	46 (100%)	0
Single/complex beats	22 (47.82%)/8 (17.34%)	0

**Table 3 ijerph-18-12194-t003:** The characteristics of QT, QTp, and TpTe intervals in the group of children with ventricular arrhythmia depending on the T-wave morphology.

Children with Ventricular Arrhythmias					Mann-Whitney U Test *p*-Value
T-Wave Normal			T-Wave Abnormal		
N	36		10		
Parameter [ms]	Sum of Ranks	Mean of Ranks	Sum of Ranks	Mean of Ranks	
**QT**	824.0	22.9	257.0	25.7	0.562
**QTp**	844.5	23.5	236.5	23.7	0.978
**TpTe**	775.5	21.5	305.5	30.6	0.033

**Table 4 ijerph-18-12194-t004:** The characteristics of QTcB and QTcF intervals in the group of children with arrhythmia ventricular depending on the T-wave morphology.

Children with Ventricular Arrhythmias					Student’s *t*-Test *p*-Value
T-Wave Normal			T-Wave Abnormal		
N	36		10		
Parameter [ms]	Mean	Standard Deviation	Mean	Standard Deviation	
**QTcB**	409.1	24.9	423.1	28.3	0.134
**QTcF**	394.9	16.9	405.7	21.4	0.097

**Table 5 ijerph-18-12194-t005:** The characteristics of QT, QTp, and TpTe intervals in the group of children with ventricular arrhythmia with T-wave abnormal and children from the control group with T-wave normal.

	Children with Ventricular Arrhythmias		Children from the Control Group		Mann-Whitney U Test *p*-Value
	T-Wave Abnormal		T-Wave Normal		
N	10		34		
Parameter [ms]	Sum of Ranks	Mean of Ranks	Sum of Ranks	Mean of Ranks	
**QT**	288.0	28.8	702.0	20.6	0.080
**QTp**	206.0	20.6	784.0	23.1	0.610
**TpTe**	395.0	39.5	595.0	17.5	<0.001

**Table 6 ijerph-18-12194-t006:** The characteristics of QTcB and QTcF intervals in the group of children with ventricular arrhythmia with T-wave abnormal and children from the control group with T-wave normal.

	Children with Ventricular Arrhythmias		Children from the Control Group		Student’s *t*-Test *p*-Value
	T-Wave Abnormal		T-Wave Normal		
N	10		34		
Parameter [ms]	Sum of Ranks	Mean of Ranks	Sum of Ranks	Mean of Ranks	
**QTcB**	423.1	28.3	391.1	21.3	<0.001
**QTcF**	405.7	21.4	379.4	18.1	<0.001

**Table 7 ijerph-18-12194-t007:** The interval between QT, QTp, and TpTe in the group of children with ventricular arrhythmia with clinical symptoms and without them.

	Children with Ventricular Arrhythmias				Mann-Whitney U Test *p*-Value
	Children with Clinical Symptoms		Children without Clinical Symptoms		
N	28		18		
Parameter [ms]	Sum of Ranks	Mean of Ranks	Sum of Ranks	Mean of Ranks	
**QT**	678.5	24.2	402.5	22.4	0.653
**QTp**	708.5	25.3	372.5	20.7	0.260
**TpTe**	554.5	19.8	526.5	29.3	0.020

**Table 8 ijerph-18-12194-t008:** The interval between QTcB and QTcF in the group of children with ventricular arrhythmia with clinical symptoms and without them.

	Children with Ventricular Arrhythmias				Student’s *t*-Test *p*-Value
	Children with Clinical Symptoms		Children without Clinical Symptoms		
N	28		18		
Parameter [ms]	Mean	Standard Deviation	Mean	Standard Deviation	
**QTcB**	410.6	27.1	414.6	24.8	0.615
**QTcF**	396.8	19.3	397.9	17.0	0.850

**Table 9 ijerph-18-12194-t009:** The characteristics of TpTe interval in the group of children with ventricular extrasystoles recorded in standard ECG from the children group with ventricular arrhythmias recorded in 24 h Holter ECG monitoring.

	Children with Ventricular Arrhythmias				Mann-Whitney U Test *p*-Value
	Ventricular Extrasystoles Recorded in Standard ECG		Ventricular Extrasystoles Recorded in Holter ECG		
N	12		34		
Parameter	Sum of Ranks	Mean of Ranks	Sum of Ranks	Mean of Ranks	
TpTe [ms]	410.0	34.2	671.0	19.7	<0.001
Number of extrasystoles recorded in 24-h Holter ECG	360.0	30.0	721.0	21.2	0.035

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
