# Peer review of "Diagnostic Value of the TpTe Interval in Children with Ventricular Arrhythmias"

_ijerph, 2021, doi:10.3390/ijerph182212194_

Round 1

Reviewer 1 Report

The authors found frequently prolonged TpTe and abnormal T-waves in children with ventricular arrhythmias, but not in those without, showed a correlation between abnormal T waves and prolonged TpTe, and noted that in the asymptomatic patients with arrhythmia, QTc was usually normal while TpTe was often abnormal.  They recommended evaluation of the T-wave and measurement to TpTe in children with arrhythmias to detect “increased arrhythmogenicity.”  However, it is unclear to me how the investigators drew the conclusion that conclusion, as they did not describe levels or categories of arrhythmogenicity, nor did the conduct a follow up to identify greater risk.

It is not clear how the subjects for this study were selected.  More information about how they were identified is needed, including the control group.

Why did the investigators not include an evaluation of JTp (J point to peak of the T-wave), which has been shown to be highly predictive of arrhythmia risk status?

The descriptions and words used to identify abnormal T-waves were not defined and not words usually used to describe T-waves in English (such as “Bactrian” and “humid”).  I have assumed they are considering T-waves with two peaks and biphasic T-waves as abnormal.  Did they not consider flat t-waves to be abnormal?  If not, why not?

When there were two T-wave peaks, which one was used for TpTe measurement.  This has been shown to be important in recent studies.

Leads II and V5 were measured.  Which lead was used to determine intervals?  Or were the measurements averaged>  How many readers did the measurements.  Was uniformity among the readers tested?

Is there an error in Figure 3 (no box for plot on the right)?

Reviewer 2 Report

I read with interest the study about the relation between depolarization characteristics in the ECG and ventricular arrhythmia in children. Nevertheless I do not find much novelty in the study and several crucial data are missing, which limits the value of the results and conclusions to the clinical practice. These are:

  1. Was the arrhythmia idiopathic? This statement is suggested by the inclusion criteria, but not mentioned in the title or text.
  2. Data on PVCs characteristics are missing: frequency/24h, presence/absence of complex arrhythmia, arrhythmia morphology (origin) - all these information are crucial in the diagnostics process, but were not taken into consideration
  3. Was cardiac MR done to exclude LV scars and myocardial diseases, also was maximal exercise done to examine the response of arrhythmia to exercise.
  4. So far, we can only find out that analyzed repolarization parameters are different in children with incorrect and correct T-waves, which is intuitive and that TpTe interval is found more frequently in children with various (also benign) symptoms, which does not necessarily has to express the severity of the arrhythmia and therefore is not either a diagnostic or prognostics factor.
  5. There are several typographical mistakes, the whole manuscript seems like a work in progress. 
  6. References are not up-to-date with many positions published in Polish language only.  

Round 2

Reviewer 1 Report

No further comments

Author Response

Proszę zobaczyć załącznik. 

Reviewer 2 Report

None, the responses do not improve the manuscript significantly to change the overall low value of this data to either scientific or clinical practice. I therefore cannot change my opinion.

Author Response

Proszę zobaczyć załącznik. 

This manuscript is a resubmission of an earlier submission. The following is a list of the peer review reports and author responses from that submission.

Round 1

Reviewer 1 Report

I don't think the statistc analyses which the authors have done are adequate for instance the analysis for normal distribution and non-normal distribustion are mixed. Further more, there are some analyses which have no use to be done, i.e. table 3., table 5. and table. 7.  And this article adds no new insights regarding to the topic. 

Reviewer 2 Report

The authors provide findings interpreted to indicate repolarization abnormalities and differences among children with ventricular arrhythmias and normal children.

The age range in the study is very broad: 4 to 18 years.  The normal ECG within this range varies extensively with age, and gender is also influential.  Without age- and gender-matching of control subjects to the arrhythmia subjects, the reported results are unreliable, as they may be the result of differences in the age distribution (not reported) and sex (53.7% vs. 65%).

ECG reading was not adequately described.  Did one reader read all of the tracings?  If not, could differences between the two groups be explained by differences in the readers(a well-known factor in ECG reading).  Were readings confirmed by second or third readers?

Biphasic T-waves were present.  In these cases, which T-wave peak was used to calculate TpTe?  Generally, it is not wise to use V% for ECG measurements, because placement location of this electrode is subject to variation.  In addition, the abnormal T-wave morphology present in V5 in some cases could have made the fiducial points harder to discern, and that could be the cause for observed group differences.